# Machine Learning-Based Classification of Transcriptome Signatures of Non-Ulcerative Bladder Pain Syndrome

**DOI:** 10.3390/ijms25031568

**Published:** 2024-01-26

**Authors:** Akshay Akshay, Mustafa Besic, Annette Kuhn, Fiona C. Burkhard, Alex Bigger-Allen, Rosalyn M. Adam, Katia Monastyrskaya, Ali Hashemi Gheinani

**Affiliations:** 1Functional Urology Research Laboratory, Department for BioMedical Research DBMR, University of Bern, 3008 Bern, Switzerland; akshay.akshay@unibe.ch (A.A.); mustafa.besic@unibe.ch (M.B.); fiona.burkhard@insel.ch (F.C.B.); 2Graduate School for Cellular and Biomedical Sciences, University of Bern, 3012 Bern, Switzerland; 3Department of Gynaecology, Inselspital University Hospital, 3010 Bern, Switzerland; annette.kuhn@insel.ch; 4Department of Urology, Inselspital University Hospital, University of Bern, 3012 Bern, Switzerland; 5Urological Diseases Research Center, Boston Children’s Hospital, Boston, MA 02115, USA; aab589@g.harvard.edu (A.B.-A.); rosalyn.adam@childrens.harvard.edu (R.M.A.); 6Department of Surgery, Harvard Medical School, Boston, MA 02114, USA; 7Broad Institute of MIT and Harvard, Cambridge, MA 02142, USA

**Keywords:** machine learning, performance, bladder, pain, gene signature

## Abstract

Lower urinary tract dysfunction (LUTD) presents a global health challenge with symptoms impacting a substantial percentage of the population. The absence of reliable biomarkers complicates the accurate classification of LUTD subtypes with shared symptoms such as non-ulcerative Bladder Pain Syndrome (BPS) and overactive bladder caused by bladder outlet obstruction with Detrusor Overactivity (DO). This study introduces a machine learning (ML)-based approach for the identification of mRNA signatures specific to non-ulcerative BPS. Using next-generation sequencing (NGS) transcriptome data from bladder biopsies of patients with BPS, benign prostatic obstruction with DO, and controls, our statistical approach successfully identified 13 candidate genes capable of discerning BPS from control and DO patients. This set was validated using Quantitative Polymerase Chain Reaction (QPCR) in a larger patient cohort. To confirm our findings, we applied both supervised and unsupervised ML approaches to the QPCR dataset. A three-mRNA signature TPPP3, FAT1, and NCALD, emerged as a robust classifier for non-ulcerative BPS. The ML-based framework used to define BPS classifiers establishes a solid foundation for comprehending the gene expression changes in the bladder during BPS and serves as a valuable resource and methodology for advancing signature identification in other fields. The proposed ML pipeline demonstrates its efficacy in handling challenges associated with limited sample sizes, offering a promising avenue for applications in similar domains.

## 1. Introduction

Lower urinary tract dysfunction (LUTD) often arises from diverse etiologies, giving rise to a spectrum of symptoms primarily centered around the bladder. These symptoms include pain, frequency, urgency, urinary incontinence, slow stream, hesitancy, and incomplete emptying. Astonishingly, an estimated 45.2% of the global population experiences at least one LUT symptom [1]. Bladder dysfunction causes a significant socioeconomic burden: in the USA, the healthcare costs of Overactive Bladder Syndrome (OAB) patients were more than 2.5 times those of similar patients without OAB [2]. Many symptoms of interstitial cystitis, or bladder pain syndrome (BPS/IC), including daytime frequency, nocturia, and urgency, are shared with OAB. Similar to OAB, BPS patients report a considerably worsened overall quality of life [3], with serious implications for mental health [4,5,6]. Bladder outlet obstruction due to the benign prostatic hyperplasia is a leading cause of OAB in ageing males [7]. Patients with benign prostatic obstruction (BPO) often present with detrusor overactivity (DO), diagnosed as involuntary spontaneous bladder contractions during filling, which can induce the symptoms of OAB [8,9].

Changes of bladder function during LUTD are the manifestation and the consequence of the alterations in bladder morphology, making the identification of genetic biomarkers of bladder function an attractive possibility to improve diagnosis [10]. Previously, using NGS and transcriptome analysis of the biopsies of patients with BPO, we identified gene expression profiles correlating with different bladder functional phenotypes (bladder outlet obstruction with or without DO, and underactive acontractile bladders), and delineated activated signalling pathways and contributing regulatory miRNAs [11]. Recently, we carried out an integrated mRNA–miRNA transcriptome analysis of the bladder biopsies from patients with non-ulcerative BPS and identified signalling alterations contributing to the disease pathogenesis [12]. These studies lay the foundation for identifying molecular markers that can effectively classify various types of LUTD by leveraging NGS datasets.

In this context, here we introduce a robust framework utilizing machine learning (ML) for the discovery of potential biomarkers, aiding in the identification of molecular drivers in BPS. We first validated our prior NGS findings related to two LUTD conditions, non-ulcerative BPS and DO, within a larger and independent patient cohort, utilizing a Quantitative Polymerase Chain Reaction (QPCR). To discern potential biomarkers from the validated gene set, we applied a combination of unsupervised and supervised ML techniques. The objective was to identify biomarkers capable of distinguishing between BPS and BPO with DO based on their distinctive transcriptomic profiles.

## 2. Results

### 2.1. Transcriptomic Difference between BPS and DO

#### 2.1.1. Differential Expression Analysis

BPS and DO share many symptoms, potentially indicative of common pathways mediating bladder sensation [13]. To understand gene expression patterns underlying these conditions, we carried out comparative NGS analyses in both patients’ groups using bladder dome biopsies, and compared them to controls [11,12].

Both DESeq2 and edgeR algorithms were applied to conduct differential expression analysis. Utilizing a set of significantly differentially expressed genes (DEGs) identified in patients with BPS (*n* = 6) and BPO with DO (*n* = 6) in comparison to control patients’ (*n* = 6), hierarchical clustering was performed. The patients with BPS form a separate cluster from those with DO, indicating substantial disparities in gene expression profiles between the two groups (Figure 1A).

Further assessment utilizing a similarity matrix based on the DEGs demonstrated a higher degree of homogeneity within the BPS patient group compared to the DO group (Appendix A). Among the 980 DEGs identified in the BPS dataset and the 1203 DEGs in the DO dataset, as determined by DESeq2 in comparison to the control group, a mere 69 DEGs were shared (Figure 1B). Notably, of these shared DEGs, only 21 genes demonstrated a common regulatory pattern across both BPS and DO datasets. Additionally, 20 genes exhibited opposite regulatory patterns, with some being up-regulated in BPS and down-regulated in DO, or vice versa (Figure 1C). These observations highlight the transcriptomic-level differences between patients with BPS and those with DO.

#### 2.1.2. Functional Enrichment Analysis

Biological function analysis [11], based upon differentially regulated pathways identified through Ingenuity Pathway Analysis (IPA), revealed fewer enriched pathways in BPS (Figure 1D) compared to DO (Figure 1E). Interestingly, metabolic pathways were activated in both types of LUTD. In BPS, there was a higher prediction of enriched biological functions related to cellular stress and injury, as well as intracellular and second messenger signalling, compared to DO (Figure 1D,E). Overall, the biological functions of 185 significant pathways in DO and the 45 pathways predicted in BPS revealed a minimal overlap (Figure 1D,E), underscoring the functional distinctions between BPS and DO.

Utilizing DEGs for both BPS and DO groups, we conducted Gene Ontology (GO) Over Representation Analysis (ORA) and gene set enrichment analysis (GO-GSEA) in both cohorts (Appendix A). Notably, GO-GSEA identified a substantially higher number of enriched gene sets in DO compared to BPS, indicative of a broader spectrum of dysregulated processes. Inflammatory responses and extracellular matrix remodelling were the hallmarks of DO (Appendix A), whereas cell division and regulation of nervous system development were highly represented in the BPS dataset (Appendix A). A noteworthy aspect of the BPS dataset was the prominence of peripheral nervous system development. Both types of LUTD shared processes including muscle contractility, cell proliferation, immune response, and neuronal activation.

### 2.2. Validation of RNA-Seq Data Using QPCR

#### 2.2.1. QPCR Gene Panel Selection Criteria

To curate a gene panel capable of effectively discerning between patients with DO and BPS, we systematically identified genes exclusively regulated in BPS (with a log2 fold change > ± 0.5 and a *p*-value < 0.05). This selection process yielded 12 genes: TPPP3, FAT1, SMTN, ANGPTL7, CLEC3B, AIM1, PALM, NCALD, P2RX2, NRXN2, FAM83A, and MFAP5 (Figure 1F). Subsequently, we incorporated NRXN3, a gene exclusively regulated in DO and not in the BPS dataset (Figure 1G), resulting in a comprehensive 13-gene panel for subsequent investigations (Figure 1H).

#### 2.2.2. Statistical Examination of Selected Genes

To assess the suitability of the chosen 13-gene panel, we employed the following statistical tests.

Normality Test on RNA Seq Data: According to the central limit theorem, the sampling distribution tends to be normal if the sample is large enough (*n* > 30). However, our sample size for RNAseq is smaller (*n* = 6); therefore, normality was checked by visual inspection [histogram plots, Q–Q plot (quantile-quantile plot)] and by significance tests. Using normalized read counts in histogram plots, Q–Q plots, and the Shapiro–Wilk normality test, we established that the distribution of the data was significantly different from normal (Appendix A). Based on read counts in all groups (BPS, DO, and control), we visualized regulation for each of the selected genes in the NGS dataset (Figure 2). After performing the Kruskal–Wallis test, we showed that TPPP3, SMTN, ANDPTL7, NCALD, and P2RX2 are up-regulated in BPS compared to control and DO; FAT1, AIM1, and FAM83A are down-regulated; CLEC3B and PALM are higher in BPS than in DO; and NRXN3 is up-regulated in DO compared to BPS and control (Figure 2).

Empirical Cumulative Distribution Function (ECDF) Analysis: ECDF is closely related to cumulative frequency and provides an alternative visualization of distribution. It reports for any given number the percentage of individuals that are below a set threshold. We applied this function to NGS read count data for the 13 selected marker genes and report an excellent separation in distribution of reads between BPS, DO, and controls for some genes (TPPP3, FAT1, ANGPTL7, AIM1, PALM, NCALD, P2RX2), but not others (SMTN, NRXN3, NRXN2, FAM83A, MFAP5) (Figure 3).

Z-score-based Patient Grouping: To assess the potential utility of the 13 chosen genes for categorizing patients based on their LUTD type, we computed a patient z-score for each gene. This score represents the deviation of an expression value from the mean expression of that gene across all patients. Patients were then categorized into groups based on their calculated z-scores. A patient with a z-score >0 was classified as High (indicated by red bars in Appendix A), while a z-score <0 designated the patient as Low (indicated by green bars in Appendix A). Upon comparing z-score for patients with DO and BPS, it becomes evident that the selected genes effectively segregate the samples based on the type of LUTD. High z-scores are observed in BPS patients for TPPP3, ANGPTL7, CLEC3B, PALM, NCALD, and P2RX2. Conversely, low z-scores are observed in BPS patients for FAT1, AIM1, and NRXN3. Z-scores do not effectively distinguish between BPS and DO groups based on SMTN, FAM83A, and MFAP5 genes (Appendix A).Correlation Analysis: We used a correlogram to determine the relationship between different attributes (genes). Appendix A shows correlation with significance values added. FAM83A, FAT1, and AM1 showed an opposite relationship or non-significant correlation to other genes, whereas the rest of the selected genes had a positive correlation to other genes. In particular, MFAP5 showed strong correlation to CLEC3B, PALM, SMTN, TPPP3, and NCALD (Appendix A).Principal Component Analysis (PCA): We conducted principal component analysis (PCA) on 18 patients, divided into three groups (BPS, DO, and control, with *n* = 6 in each group). The analysis was based on the NGS read counts of 13 selected genes. PCA affirmed that these 13 genes have the capability to distinctly cluster all patients according to their LUTD type (Figure 4A). The scree plot further illustrates that PC1 captured 59.9% of the total variance, while PC2 captured 13.7% of the variance (Appendix A). Among the 13 genes, MFAP5, PALM, NCALD, TPPP3, and CLEC3B significantly contributed to PC1 (over 10% each), while NRXN3, FAM83A, P2RX2, and ANGPTL7 were the primary contributors to PC2 (Appendix A).Clustering Analysis: The hierarchical clustering algorithm was applied to the Next-Generation Sequencing (NGS) read counts of 13 selected genes. The resulting tree was divided into *k* clusters. We specifically investigated whether setting *k* = 3, corresponding to the known groups (BPS, DO, and Control), accurately represented the grouping. The hierarchical cluster dendrogram grouped 18 samples into three distinct clusters: one comprising all BPS samples and one DO sample, another consisting solely of DO samples, and a third including all control samples along with two DO samples (Figure 4B). Furthermore, we utilized both the Elbow method (Figure 4C) and Clustree (Figure 4D) to determine the optimal number of clusters for the given datasets. Both methods consistently identified *n* = 3 as the optimal number of clusters. We delved deeper into sample clustering using the *k*-means partitioning clustering method (Figure 4E). Our observations revealed that setting *k* = 3 effectively separated NGS patients into three groups. Specifically, DO1, DO4, and DO3 clustered together, while DO2, DO5, and DO6 clustered with control samples. Notably, all BPS patients formed a distinct cluster, separate from both control and DO patients.

#### 2.2.3. QPCR Validation of Selected Genes in a Larger Patient Cohort

Using an independent cohort of controls (*n* = 14), patients with BPS (*n* = 28), and BPO patients with DO (*n* = 22), we carried out QPCR analysis of the levels of mRNA for the 13-gene panel described above. In the larger sample group, the log2 fold change (log2FC) values showed normal distribution, confirmed by the Shapiro—Wilk test (Appendix A). We compared the expression levels of each gene in the patient groups and showed that TPPP3, SMTN, ANGPTL7, CLEC3B, PALM, NCALD, P2RX2, and NRXN2 are significantly up-regulated in BPS compared to control and DO; FAT1 was up-regulated in BPS and down-regulated in DO compared to control; AIM1 was down-regulated in both DO and BPS compared to controls; and FAM83A was significantly down-regulated in BPS (Appendix A). The empirical cumulative distribution function (ECDF) showed good separation in the distribution of reads between BPS, controls, and DO for TPPP3, NCALD, SMTN, NRXN2, and FAT1 (Appendix A). We used Deviation Graphs to visualize the calculated z-score for the deviation of quantitative values from a reference value (mean of controls) for each gene expression log2FC value. Z-scores were high for BPS and low for DO in TPPP3, FAT1, SMTN, ANGPTL7, CLEC3B, PALM, NCALD, and P2RX2; z-scores did not separate groups in AIM1, NRXN3, FAM83A, and MFAP5 (Appendix A). The correlogram for the selected genes validated by PCR is shown in Appendix A. PCA using log2FC of QPCR data showed that PC1 captured 48.8% of all variances (Appendix A). NCALD, TPPP3, ANGPTL7, and SMTN strongly contributed to PC1 (over 10%), while AIM1, FAT1, and P2RX2 were the main contributors to PC2 (Appendix A). Sample distribution in PCA (Appendix A) shows that log2FC values of the 13 marker genes effectively separated BPS from DO with control samples in between.

### 2.3. Identification of BPS mRNA Signatures through Unsupervised ML Analysis of QPCR Data

We initiated our analysis with an exploratory examination, utilizing hierarchical clustering based on QPCR data. The results revealed a distinct separation of BPS samples from both DO and control groups. Intriguingly, some DO samples co-clustered with controls, while others formed a distinct subgroup (Figure 5A). To ascertain the optimal number of clusters within the QPCR dataset, we applied the Elbow method (Figure 5B) and Clustree analysis (Figure 5C), with both methodologies consistently identifying three as the optimal number of clusters.

Subsequently, employing the *k*-means algorithm with *k* = 3 reaffirmed the efficacy of sample partitioning (Figure 5D). Within the *k*-means clustering, the top two contributors to variance in Dimensions 1 and 2 were identified as NCALD, TPPP3, AIM1, and FAT1 (Figure 5E). Notably, during this clustering process, Cluster 3 (depicted in green, (Figure 5D)), comprising the majority of BPS samples, exhibited elevated expression levels of NCALD, FAT1, and TPPP3. This observation suggests the potential of these genes to serve as mRNA markers for BPS.

### 2.4. Supervised Machine Learning

#### 2.4.1. Addressing Imbalanced Small Sample Sizes in Multi-Class Classification

In the current QPCR dataset, we have samples from three different groups: controls, BPS, and DO, making it a multi-class classification problem. While most ML algorithms were originally designed for binary classification [14], they can be adapted for multi-class problems using strategies like One-vs-Rest and One-vs-One. In our case, we employed the One-vs-One technique, fitting one binary classification model for each pair of classes (BPS vs. Control, DO vs. Control, or BPS vs. DO).

Moreover, there is an imbalance in the number of samples within the groups in the QPCR dataset (Figure 6A). This could pose a challenge for model training as most ML classification algorithms were developed with an assumption of equal class distribution [15], and might generate irrational results with an imbalanced dataset. To address this problem, there are two common strategies: oversampling where the number of samples in the minority class is increased, or undersampling where samples from the majority class are removed [16,17]. Since our available dataset is comparatively small and the undersampling method will decrease it further, we chose oversampling for data augmentation of the minority classes. Three different oversampling methods (Borderline Synthetic Minority Oversampling Technique (SMOTE), SVM SMOTE, and Random Over Sampler) were used for the data augmentation of minority classes. In total, 36 models (12 ML algorithms × 3 data resampling methods) were trained and evaluated for every pairwise group comparison.

#### 2.4.2. ML Model Evaluation

Depending on the data distribution, any given model has its respective limitations, and because they are just estimations, none of them can be fully accurate. In order to identify biases among our tested models, we evaluated model performance by calculating seven different performance metrics (F1, Accuracy, Balanced Accuracy, Precision, Recall, Average Precision, and ROC-AUC score). However, due to lower standard deviation of F1 compared to other metrics, the F1 score has been used as a primary performance metric to sort the models (Appendix A). Nevertheless, the right choice of an evaluation metric is critical and usually depends on the problem being addressed. Therefore, we utilized an ML cumulative performance score (MLcps) based on seven different metrics to identify the best-performing ML model [18]. MLcps combines multiple evaluation metrics into a unified score and provides a comprehensive picture of different aspects of the performance of the trained model. We utilized two different model evaluation techniques, *k*-fold Cross-Validation (CV) (Appendix A) and nested CV method (Appendix A), to evaluate the performance of each model.

For BPS vs. Control, the LDA algorithm in combination with SVM SMOTE (Figure 6B and Appendix A) showed the best performance compared to other models, with an F1 score of 0.83, far superior to our baseline model, i.e., Dummy classifier (F1 score 0.55). Similarly, the best-performing models for DO vs. controls (Figure 6C and Appendix A) and BPS vs. DO (Figure 6D and Appendix A) are SVM with Borderline SMOTE (F1 score 0.87) and GP with Random Over Sampler (F1 score 0.93), respectively.
Figure 6Supervised ML model selection. (**A**) Class distribution of QPCR data. (**B**–**D**) Spider Plots. Each plot represents a ML model where the first part of the model name corresponds to a particular ML algorithm and later is a data resampling method. It displays a metrics score for the corresponding model where the surface of the shaded area reflects the performance. Here, models are sorted in increasing order based on the F1 score from the k-fold CV or Nested CV result. The three best and worst-performing models are shown for (**B**) BPS vs. Control, (**C**) DO vs. Control, and (**D**) BPS vs. DO classification problem. (**E**) 3D scatter plot of the log2FC for NCALD, FAT1, and TPPP3 in QPCR dataset. The coordinates of each sample correspond to the log2FC of the 3 mRNA markers, and the confidence ellipse represents an iso-contour of the Gaussian distribution and allows visualization of a 3D confidence interval.
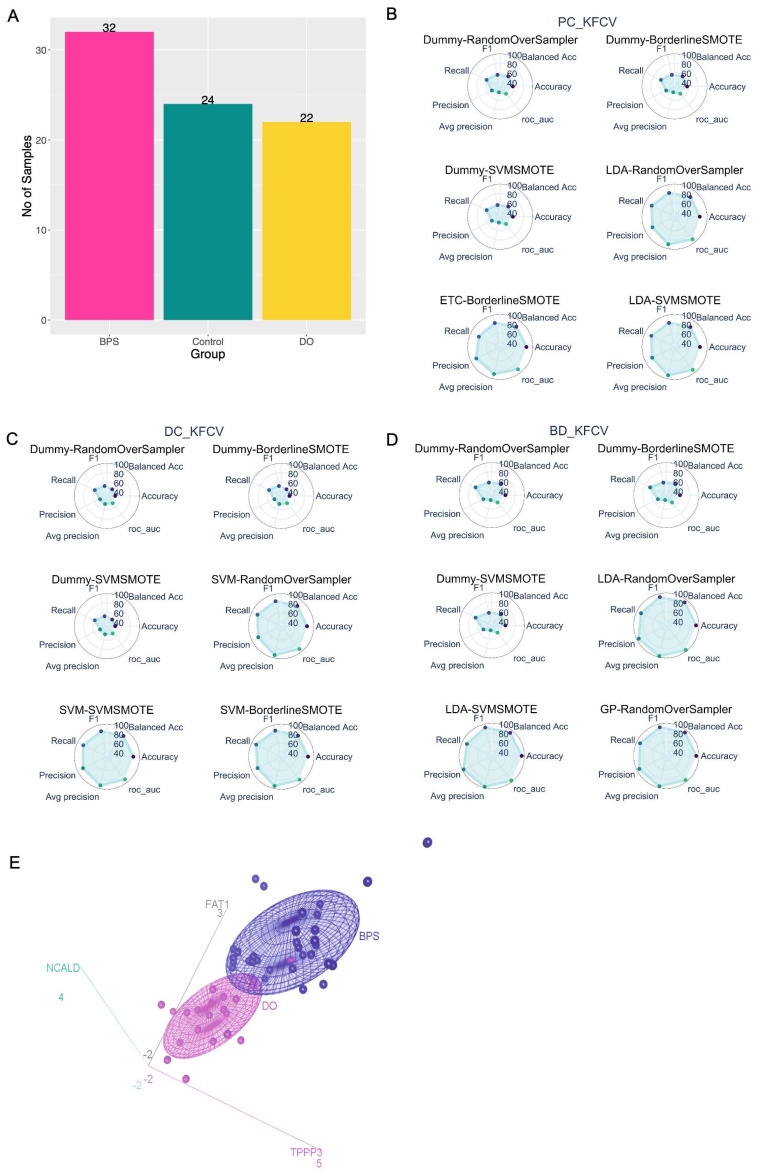


#### 2.4.3. Identification of mRNA Signatures Based on QPCR Data Using Feature Selection Technique

For each of the group pairwise comparisons, 10 out of 13 genes were selected as the most important features by the Recursive Feature Elimination (RFE) cross-validation (RFECV) method, and the remaining three genes were identified as not essential to classify either of the groups per RFECV results. Since all the models showed comparable performance, we took the intersection of selected features from each model as a final set of features, rather than features selected from the single best-performing model only. This resulted in TPPP3, FAT1, SMTN, CLEC3B, AIM1, and NCALD as a final list of selected features that could serve as potential biomarkers to differentiate BPS from controls. The number of selected features for DO vs. Control and BPS vs. DO is 5 and 6 respectively (Table 1).

#### 2.4.4. Visualization of Selected mRNA Signatures

Both supervised and unsupervised approaches commonly identified three mRNAs, NCALD, FAT1, and TPPP3, as potential signatures associated with BPS. Figure 6E presents a 3D scatter plot illustrating log2FC values for NCALD, FAT1, and TPPP3 mRNAs in bladder biopsies from 22 DO and 28 BPS patients. Each sample’s coordinates correspond to the log2FC of the three mRNA markers, and the confidence ellipse represents an iso-contour of the Gaussian distribution, facilitating the visualization of a 3D confidence interval. This plot conclusively demonstrates that these three selected mRNA signatures effectively distinguish between DO and BPS groups.

## 3. Discussion

The application of omics technologies for biomarker discovery and accurate classification of pathophysiological phenotypes often leads to extensive candidate gene lists that are unsuitable for target identification and validation. Integrating machine learning with transcriptomic approaches can be beneficial for disease classification [19,20]. However, there is no consensus on the use of machine learning algorithms to find classifiers for transcriptome data suitable for diagnosis. The diversity of biological data makes it challenging to provide all-purpose guidelines for machine learning in biology [21]. Due to the variety of input data types, the paradigm of differentiable programming is emerging from the field of deep learning [22].

LUTD encompasses syndromes with overlapping symptomatic manifestations. One of the challenges in LUTD diagnostics is the lack of robust biomarkers, reflecting the underlying origin of the disease. IC/BPS is a chronic disease of unknown etiology; its suggested causes include autoimmunity, neurogenic inflammation, and altered central nociception [23,24]. Currently available transcriptome studies in human IC/BPS patients are difficult to compare, because they were performed using different technologies and material sources. There was little consensus in the data from the gene expression microarrays in biopsies [25] and urine sediment [26]; the inflammatory mediator QPCR panel was limited to 96 genes [27], and RNA sequencing of urine sediment [28] and bladder biopsies [29] only revealed an expression profile for the ulcerative IC phenotype.

Although our earlier NGS gene expression profiling and pathway analysis shed light on the molecular changes in the bladders of patients with BPS, they did not determine the cause of the disease [12]. To reveal the molecular drivers of BPS, we employed machine learning approaches to identify a subset of DEGs that could serve as a reliable classifier, distinguishing BPS from other types of LUTD. As a proof of concept, we defined a gene signature that separated BPS and DO patients with bladder outlet obstruction.

We started with unsupervised learning methods and applied them to a small NGS dataset and a larger QPCR dataset to define gene signatures of BPS. Comparative analysis of the NGS transcriptomes of BPS and DO revealed considerable differences in the underlying biological processes: predominant activation of the immune system and inflammation were characteristic of DO, whereas cell cycle, proliferation, and regulation of the nervous system development were the hallmarks of non-ulcerative BPS. Based on the NGS data, we selected a smaller gene set (13 genes) of group-specific markers and performed unsupervised ML using their read counts. We showed that expression levels of these 13 genes segregated the BPS, control, and DO groups (*n* = 6 each) in PCA and hierarchical cluster dendrogram.

The 13 genes suggested as BPS classifiers were further validated by QPCR in larger groups of BPS (*n* = 28) and DO (*n* = 22) patients. In these sample groups, the log2 fold change expression values of the marker genes were normally distributed and correctly grouped the samples in PCA and dendrogram. When applied to *k*-means (*k* = 3), it provided good sample partitioning. Cluster 3, containing the majority of BPS samples, was characterized by an up-regulation of NCALD, FAT1, and TPPP3. Plotting log2 fold change values for NCALD, FAT1, and TPPP3 mRNAs in bladder tissues of 22 DO and 28 BPS patients, and visualization of a 3D confidence interval, showed that these three mRNAs were sufficient to discriminate DO and BPS groups from each other.

Supervised ML has emerged as a valuable tool for data interrogation and is widely used in many biological domains, such as chemoinformatics and genomics. One obstacle for ML in biology is a small number of observations in most studies. To address this, we used a QPCR validation dataset from BPS, DO, and controls to increase the number of samples available for machine learning. However, due to the limited availability of bladder biopsies, the datasets were still small. ML with comparatively small datasets can easily lead to overoptimistic results [30,31] and requires numerous sanity checks for appropriate performance metrics, evaluation methods, etc. Therefore, before selecting a particular ML algorithm, it is important to test different algorithms on a given problem and evaluate their performances from multiple perspectives as there is no general performance metric. As shown in Appendix A, the performance of a model depends on the used dataset. For example, LDA algorithms along with SVM SMOTE perform best for BPS vs. control (Appendix A) and DO vs. BPS (Appendix A), but ranked third for DO vs. control classification (Appendix A).

Therefore, instead of relying on a single method or algorithm, we propose a ML pipeline that involves evaluating multiple algorithms for each step of the ML process, such as data resampling and classification algorithms. Choosing an appropriate metric to assess the model performance is as important as choosing the machine learning algorithm. Most metrics give importance to a particular aspect of the model’s performance. For example, the recall metric can summarize how well a model can predict the positive class, but gives no information about the negative class. Therefore, the proposed pipeline calculates various metrics, providing an opportunity to evaluate multiple aspects of the trained model. These metrics are then combined into a single performance metric called MLcps, which we recently developed [18] to facilitate straightforward comparisons of trained machine learning models (Appendix A). Similarly, to avoid the overly optimistic biased evaluation of the model’s performance, the double CV or nested CV method was used for model evaluation to support the *k*-fold CV results. Nested CV performs hyperparameter tuning and model evaluation using different subsets, which, in turn, lessens, if not eradicates, the likelihood of overfitting or optimistic model evaluation [30].

To avoid external bias in feature selection, instead of manually configuring the number of features, the proposed pipeline automates the RFE method using cross-validation, where the estimator is trained with the selected features and evaluated using CV. The features that decrease or have no impact on the model performance are removed recursively until the optimal number of features is obtained. The threshold for the minimum number of features to be selected from the RFE method was three. Six features were selected for both BPS vs. control and BPS vs. DO, while five features were selected in DO vs. control. Importantly, similar to the results of the unsupervised ML, TPPP3, FAT1, and NCALD effectively segregated BPS and DO, and BPS and control.

TPPP3, FAT1, and NCALD genes are up-regulated in BPS (larger patient cohort, QPCR validation), and encode proteins detected in both human and mouse bladders. Tubulin polymerization promoting protein family member 3 (TPPP3) facilitates microtubule assembly [32] and plays a critical role in cell mitosis, one of the main activated processes according to GO ORA of the BPS dataset. Up-regulation of atypical cadherin FAT1 is an important regulator of neuromuscular morphogenesis [33], and its expression in vascular smooth muscle cells increases after injury [34]. Previously, loss of FAT1 has been linked to glomerular nephropathy [35]; however, although it was down-regulated in the NGS study, its levels were significantly higher than controls in a larger patients’ cohort, ruling out a potential link between the dysfunctions, as none of the recruited patients showed kidney deficiency. Interestingly, FAT1 was found to promote the expression of pro-inflammatory mediators and TGF-beta [36], potentially contributing to the bladder remodelling in BPS. NCALD encodes neurocalcin delta, a neuronal calcium-sensing protein, controlling clathrin-coated vesicle traffic [37]. It is a negative regulator of endocytosis, and in neurons inhibits synaptic vesicle recycling [38] and plays a role in adult neurogenesis [39]. The other features selected to discriminate BPS from DO include components of neuronal synaptic complex: paralemmin (PALM) is implicated in membrane dynamics in postsynaptic specializations, and in axonal and dendritic processes [40]. Neurexin NRXN2 promotes synapse formation [41]. Their up-regulation in BPS might be indicative of the changes in the peripheral innervation and/or the architecture of neuromuscular junctions in the detrusor.

Here we investigated whether ML methods are capable of differentiating types of LUTD with common symptoms, such as BPS and DO/OAB. Both supervised and unsupervised ML algorithms were employed to test the performance of 13 selected genes. Six features selected in supervised ML overlapped with three features revealed by unsupervised ML; the GO and biological functions of the common gene signature (TPPP3, FAT1, and NCALD) reflected the dysregulated cell proliferation, muscle contraction, and neurogenesis which might contribute to bladder pain. Our approach yielded a selection of mRNA signatures (classifiers) for BPS, effectively discriminating between BPS patients and controls, as well as other LUTD (DO).

## 4. Materials and Methods

### 4.1. Data Depository

The mRNAseq datasets were deposited in the European Nucleotide Archive (ENA). DO dataset accession numbers PRJEB11369; BPS dataset accession numbers PRJEB46961.

### 4.2. Study Approval

Studies were approved by the Ethics Committee of Canton Bern (KEK 146/05 original BPS study, KEK 331/14 follow-up BPO study), and all subjects gave written informed consent. The study was registered in ClinicalTrials.gov Protocol Record 331/2014, ClinicalTrials.gov Identifier: NCT01482676.

### 4.3. Patient Selection

Patients with DO and OAB symptomatic complex, concomitant with BPO (*n* = 22), were recruited and described in our pervious study [11]. Patient selection and evaluation were described previously [11,42]. Briefly, cold cup biopsies from the bladder dome were collected from 14 controls and 28 BPS patients in an earlier published study [42]. All subjects underwent a complete urological evaluation (including medical history, physical examination, urine culture, and flexible urethrocystoscopy). In addition, all subjects with BPO underwent uroflowmetry, post void residual (PVR), and urodynamic investigations. Patients were divided into the following groups:

(1) Group 1: Control—asymptomatic patients undergoing cystoscopy for other reasons (e.g., stent placement for stone disease, microhematuria evaluation) (*n* = 6 for NGS, *n* = 14 for QPCR validation).

(2) Group 2: BPS—patients with pain (>3 months) considered to be located in the bladder and/or frequency, urgency, and nocturia (*n* = 6 for NGS, *n* = 28 for QPCR validation).

(3) Group 3: DO—patients with bladder outlet obstruction caused by benign prostatic hyperplasia showing increased detrusor pressure and reduced urine flow during pressure flow, in combination with involuntary detrusor contractions during filling phase (phasic and/or terminal) and defined as obstructive according to the Abrams-Griffith nomogram (*n* = 6 for NGS, *n* = 22 for QPCR validation). All patients with DO had OAB.

### 4.4. Functional Enrichment Analysis

Gene Ontology (GO) over-representation analysis (ORA) [43] methods were used to gain biological insight on the DEGs. We used clusterProfiler (version 3.18.1) package [44] in R to perform GO-ORA and GO-GSEA on biological process (BP) terms associated with DEGs. Results obtained at a threshold of *p*-value below 0.1 were considered statistically significant.

### 4.5. QPCR Validation of NGS Studies

Total RNA was isolated from bladder dome biopsies using the miRVana miRNA isolation kit (Ambion, ThermoFisher Scientific, Waltham, MA, USA). The reverse transcription reactions were carried out using the High-Capacity cDNA Reverse Transcription Kit (Applied Biosystems, Waltham, MA, USA) with random hexamer primers. TaqMan assays were from Applied Biosystems. QPCR was carried out in triplicates using 7900HT Fast Real-time PCR System (Applied Biosystems, ThermoFisher Scientific, Waltham, MA, USA). The data (Ct values) were normalized to the internal reference (levels of 18S rRNA in each sample), and the log2 fold change (log2FC) related to the average value in all controls was calculated.

### 4.6. Statistics

#### 4.6.1. Hierarchical Clustering and Heatmaps

Hierarchical clustering and the associated heatmaps for miRNA and mRNA sequencing data were generated with the function heatmap2 in the R package gplots or GENE-E R package. Pairwise correlation matrix between items was computed based on the Pearson correlation method. The average linkage method used an average to calculate the distance matrix. For the heatmap visualization, the log2-expression values were used. We used dendextend R package to create and compare visually appealing tree diagrams.

#### 4.6.2. Principal Component Analysis (PCA)

‘Prcomp’ function implemented in R (R version 4.2.0 (22 April 2022 ucrt)), rgl (Version 1.2.8), and scatterplot3d (Version 0.3-44) R package were used for the principal component analysis of three-dimensional plots. The calculation was done by a singular-value decomposition of the (centered and scaled) data matrix.

For QPCR validation, the log2 fold change differences to the average of control samples were calculated. A one-way analysis of variance (ANOVA) was employed, and the Tukey correction used to correct *p* values. The *p* value < 0.05 was considered statistically significant (GraphPad Prism (version 7.01)).

#### 4.6.3. D Ellipsoid Chart and Point Identification for Biomarkers

The scatter3d() (Version 0.3-44) function in car (Companion to Applied Regression) package was used in order to call for rgl package, which draws 3D scatter plots with various regression surfaces ((R version 4.2.0 (22 April 2022 ucrt)) using XQuartz (The X Window System, version 2.7.9). The coordination of each point on the 3D graph was identified based on its Mahalanobis distance from the centroid of the three variables.

### 4.7. Deviation Graphs

The deviation graph showing the deviation of quantitative values from a reference value was visualized using a z-score calculated for each patient based on any given gene. A z-score of 1 denotes that the observation is at a distance of one standard deviation towards right from the centre. The calculation was performed in R. If z-score is >0, we call it High; if z-score is <0, we call it low.

Example:$$Z-score_{TPPP3}=\frac{readcount\left(TPPP3\right)-mean\left(TPPP3\right)\;}{sd\left(TPPP3\right)}\;\left(calculate\;for\;each\;patient\;separately\right)\;$$

### 4.8. Hierarchical Clustering Algorithm

The hierarchical clustering algorithm was performed using the function hcut() from the package factoextra, which applies hierarchical clustering and cuts the tree into k clusters.

### 4.9. Prediction of Biological Function of Canonical Pathways

We built a tool in R program that searches for involvement of each IPA canonical pathway in the Biological Function Classification Database of IPA, known as “ingenuity canonical pathway”, and counts the number of pathways involved in a specific biological function. The results are illustrated as radar graphs.

### 4.10. Running Score and Preranked List of GSEA Result

GSEA plots were created using the gseaplot2 function in enrichplot r package (R version 4.2.0 (22 April 2022 ucrt) in R.

### 4.11. Partitioning Clustering Method

To perform k-means and Partitioning Around Medoids (PAM), we specified the number of clusters to be generated (2–7). Then, we defined the total within-cluster variation, which measures to what degree the compactness of the clusters is minimized. We computed k-means in R with the kmeans() function.

### 4.12. Proposed ML Pipeline

We developed a computational framework for binary classification problems that automates several ML steps and evaluates multiple algorithms/methods for almost every step.

The proposed pipeline (Appendix A) starts with splitting the input datasets into k (3) different equal-size bins in a stratified manner, where k-1 bins will be used as training datasets and the remaining bin as a test dataset. Next, it uses a Recursive Feature Elimination (RFE) algorithm for feature selection. RFE is a wrapper-type feature selection algorithm that requires an external estimator/ML algorithm to assign coefficients or rank to features and select the most important feature based on the assigned coefficient or rank. Therefore, LR, LDA, and SVM algorithms were evaluated in the core of the RFE method, and the best-performing one was used. Next, to overcome the imbalanced dataset problem, three different oversampling methods—that is, Borderline Synthetic Minority Oversampling Technique (SMOTE), SVM SMOTE (DOI: https://doi.org/10.1613/jair.953, accessed on 18 January 2024), and Random Over Sampler—have been utilized for the data augmentation of minority classes.

In the next step, multiple ML algorithms are trained for the given problem. ML algorithms can be broadly categorized into (a) Linear Algorithms, (b) Nonlinear Algorithms, and (c) Ensemble Algorithms. Here, we have selected at least two classification algorithms from each of these categories. In total, 11 (Table 2) different classification algorithms were trained for the classification of LUTD patients. In addition, a Dummy classifier that makes predictions at random was used as a baseline to compare with other models. Overall, 36 different ML models (12 ML algorithms × 3 data resampling methods) have been trained, including the Dummy classifier. Then, the performance of the trained ML models was evaluated using the k-fold cross-validation (CV) (where *k* = 3) method which divides the whole dataset into *k* non-overlapping subsets of equal size. For each fold, (*k* − 1) subsets are used as a training dataset for the model, and the remaining subset as a test dataset to evaluate the model performance. In this way, *k* different models are trained, and the final performance of the model is estimated by taking the average of the evaluation metrics from each iteration. Since a single run of the *k*-fold cross-validation method may result in a noisy estimate of model performance, we repeated (*n*) *k*-fold CV 10 times. In addition, the nested CV method (*k* = 3, *n* = 10) has also been used for model evaluation to support the k-fold CV results. In contrast to *k*-fold CV, nested CV has two layers, i.e., inner CV, used for hyperparameter tuning, and outer CV, used for model evaluation [45]. Similar to ML algorithms, several different performance metrics—Accuracy, Balanced Accuracy, Precision, Recall, Average Precision, and ROC-AUC score—have been used to evaluate the performance of ML models.

The complete pipeline was developed on top of the scikit-learn (version 0.24.1) [46] library and imblearn (version 0.8.0) [47] from Python (version 3.8.8) [48]. Pandas (version 1.2.4) was used to store and process the data. Plotly (version 5.3.1), ggplot2 (version 3.3.5), and circlize (version 0.4.13) [49] were used to generate the figures.

## 5. Conclusions

We established a ML framework for analyzing NGS and QPCR-derived gene expression datasets from a limited number of samples, enabling the identification of robust classifiers for non-ulcerative BPS. The proposed pipeline incorporates various algorithms for data resampling and classification, utilizing a nested CV approach to prevent overfitting and ensure a reliable model evaluation. This comprehensive strategy enhances the robustness of the identified classifiers, making the pipeline a valuable tool for unraveling complex molecular signatures in bladder disorders while mitigating potential issues related to ML.

## Figures and Tables

**Figure 1 ijms-25-01568-f001:**
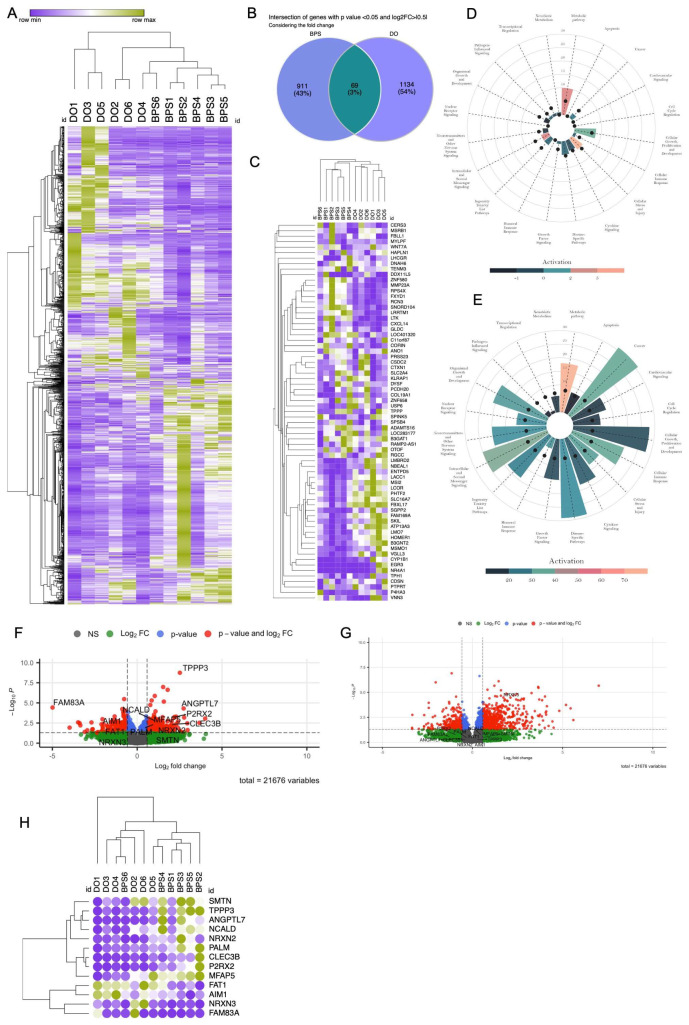
Differentially expressed genes in BPS compared to DO. (**A**) Hierarchical clustering and heatmap of normalized read counts of all significantly regulated mRNA (1390 mRNAs, *y* axis) from patients with BPS (“BPS1-6”) and BPO patients with DO (“DO1-6”) in 12 patients (*x* axis). Clustering metric is one minus Pearson correlation and linkage method is average. A relative color scheme has been used by taking min and max values in each row to convert values into colors. (**B**) Venn diagram of differentially expressed genes (DEGs) in BPS and DO patients. (**C**) Hierarchical clustering and heatmap of log2 fold changes (log2FC) of 69 common DEGs between BPS and DO patients compared to control. (**D**) Functional enrichment analysis based on IPA pathways, BPS dataset. Z-score (activation or inhibition status) is shown as a bar (**E**) Functional enrichment analysis based on IPA pathways, DO dataset. Z-score (activation or inhibition status) is shown as a bar (**F**) Volcano plot of BPS compared to the control group annotated with 13 selected genes. (**G**) Volcano plot of DO compared to the control group annotated with 13 selected genes. (**H**) Hierarchical clustering and heatmap of normalized read counts of the 13 selected genes.

**Figure 2 ijms-25-01568-f002:**
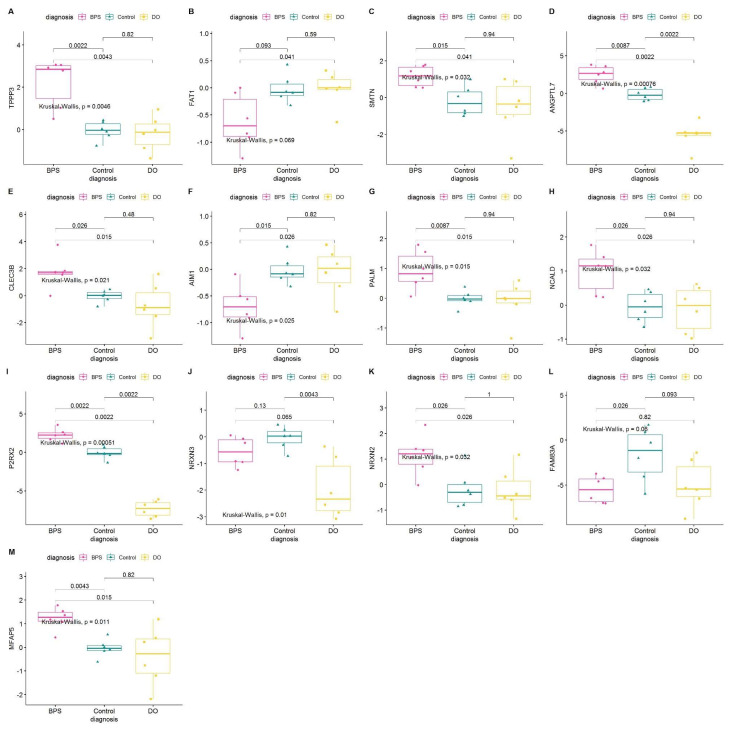
Differential Expression of selected 13 markers in NGS datasets of BPS, DO, and controls. Boxplot Statistics visualization (based on log2FC), pairwise comparisons *p*-value. (**A**) TPPP3 (**B**) FAT1 (**C**) SMTN (**D**) ANGPTL7 (**E**) CLEC3B (**F**) AIM1 (**G**) PALM (**H**) NCALD (**I**) P2RX2 (**J**) NRXN3 (**K**) NRXN2 (**L**) FAM83A (**M**) MFAP5.

**Figure 3 ijms-25-01568-f003:**
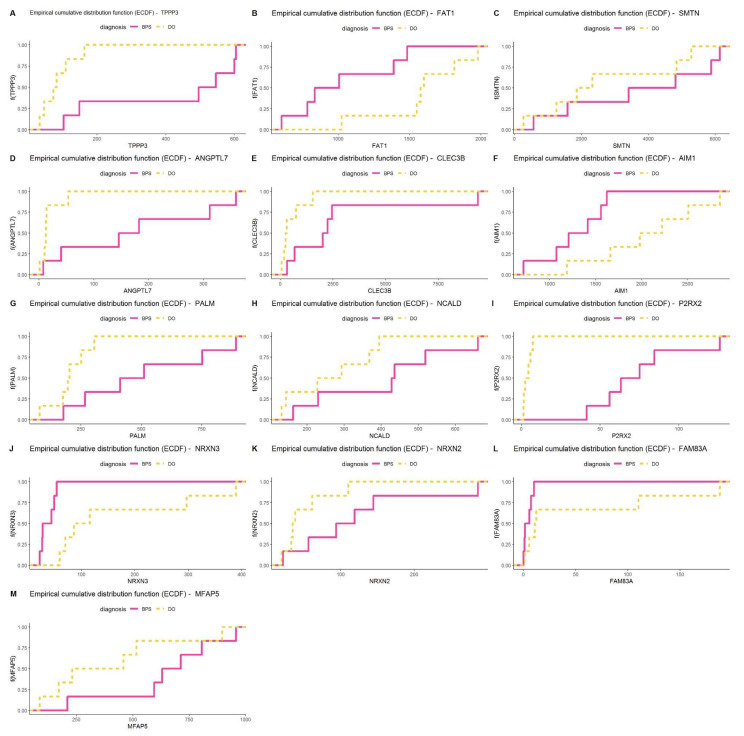
Empirical cumulative distribution function (ECDF) for the performance selected 13 markers in NGS datasets of BPS and DO. The ECDF, or empirical cumulative distribution function, reports for any given number (mRNA read count) the percentage of individuals that are below a set threshold. (**A**) TPPP3 (**B**) FAT1 (**C**) SMTN (**D**) ANGPTL7 (**E**) CLEC3B (**F**) AIM1 (**G**) PALM (**H**) NCALD (**I**) P2RX2 (**J**) NRXN3 (**K**) NRXN2 (**L**) FAM83A (**M**) MFAP5.

**Figure 4 ijms-25-01568-f004:**
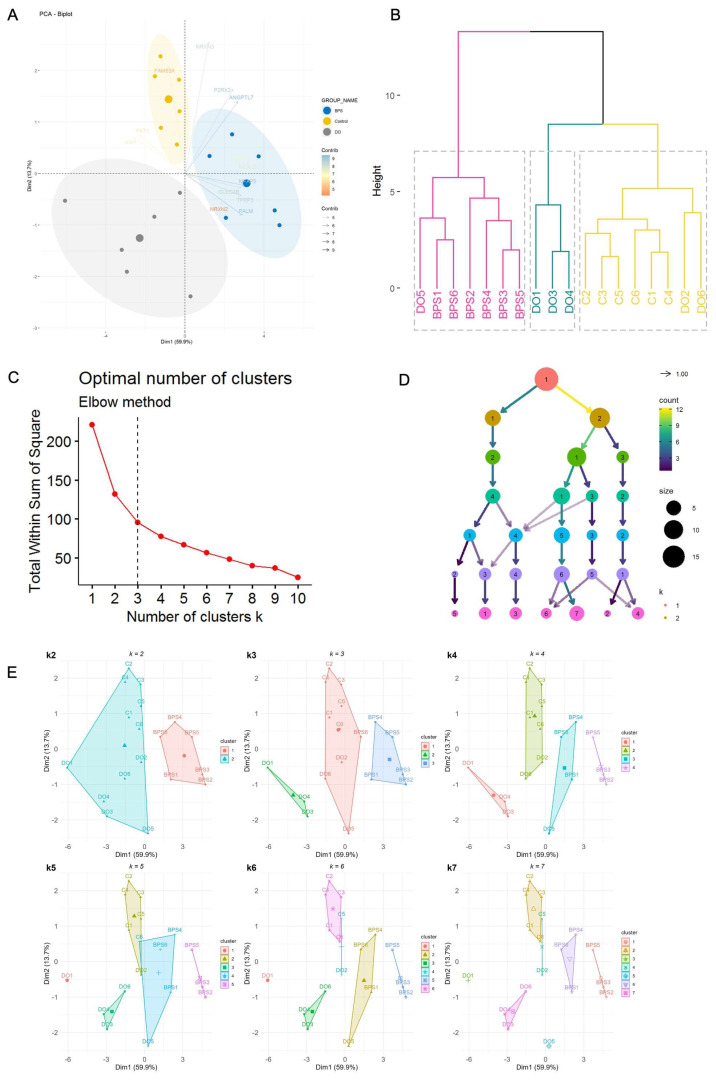
Clustering and unsupervised ML on NGS data using 13 selected genes, selecting *k* = 3 based on patients’ clustering. (**A**) PCA analysis based on the NGS read counts of 13 selected genes. (**B**) Hierarchical clustering of samples using read counts of 13 selected genes. (**C**) Optimal number of clusters for k-means by Elbow method. A change of slope from steep to shallow (an elbow) is used to determine the optimal number of clusters. (**D**) The Clustree method depicts how samples change groupings as the number of clusters increases. The size of each node corresponds to the number of samples in each cluster, and the arrows are colored according to the number of samples each cluster receives. The numbers inside the circles refer to the cluster consecutive number (1 = 1st, 2 = 2nd, …) (**E**) Partitioning clustering methods.

**Figure 5 ijms-25-01568-f005:**
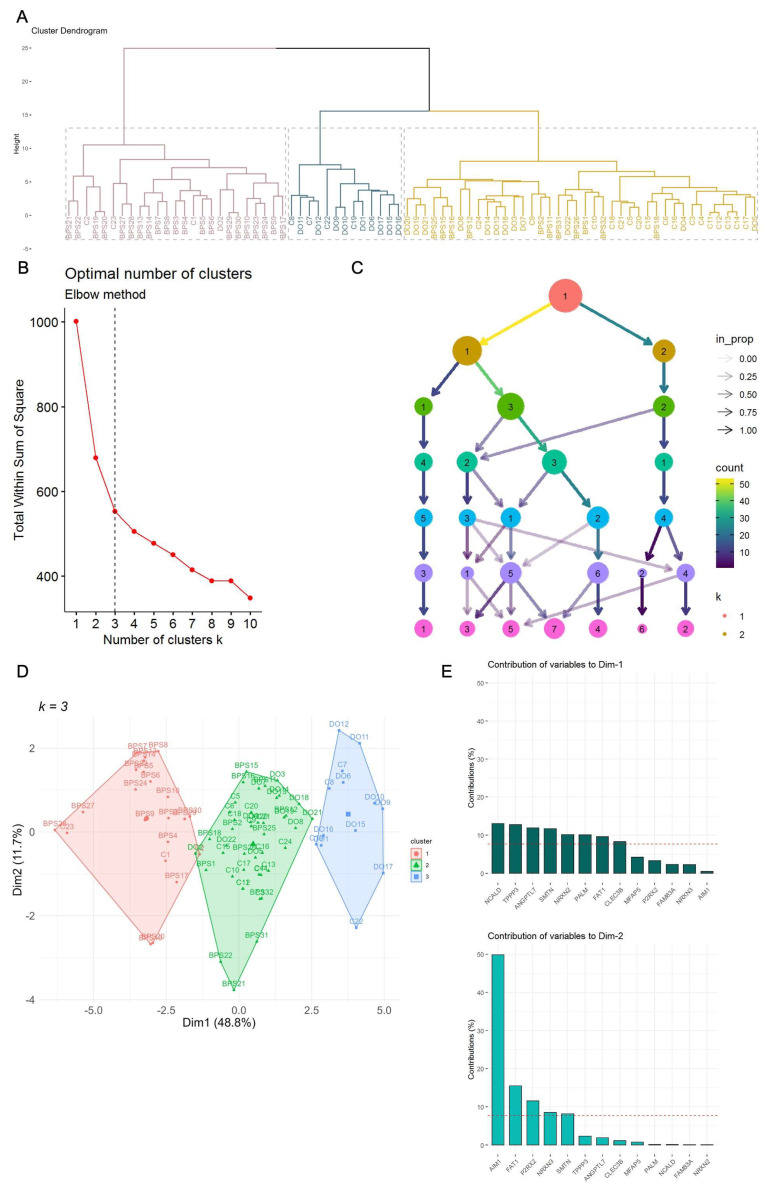
Unsupervised ML on QPCR dataset. (**A**) Hierarchical clustering on QPCR dataset. (**B**) Optimal number of clusters for k-means (Elbow method). The bend indicates that additional clusters beyond the third have little value. (**C**) The Clustree depicting how samples change groupings as the number of clusters increases. (**D**) Partitioning clustering method. (**E**) The percentage of each gene’s contribution to each dimension. A reference dashed line is also shown on the barplots corresponding to the expected value if the contribution where uniform.

**Table 1 ijms-25-01568-t001:** The table below represents the genes selected using the Recursive Feature Elimination with Cross-Validation (RFECV) method in the supervised machine learning pipeline. These identified genes can effectively differentiate between specified groups based on their expression profiles.

BPS vs. Control	DO vs. Control	BPS vs. DO
TPPP3	TPPP3	TPPP3
FAT1	CLEC3B	FAT1
SMTN	AIM1	PALM
CLEC3B	P2RX2	NCALD
AIM1	NRXN2	NRXN2
NCALD		FAM83A

**Table 2 ijms-25-01568-t002:** ML algorithms tested for performance in this study.

Classification Algorithm	Abbreviation
Logistic Regression	LR
Linear Discriminant Analysis	LDR
Gaussian Naive Bayes	GNB
Support Vector Machine	SVM
k-nearest neighbors	KNN
Decision Tree Classifier	DTC
Gaussian Process Classifier	GP
Random Forest Classifier	RF
Bagging Classifier	BC
Extra Trees Classifier	ETC
Gradient Boosting Classifier	GBC

## Data Availability

The mRNA- and miRNA-seq datasets were deposited in the European Nucleotide Archive (ENA) under ENA accession numbers: PRJEB46961 for mRNA and PRJEB10955 for miRNA. The machine learning pipeline used in this project is available on github (https://github.com/FunctionalUrology/MLcps) (accessed on 18 January 2024) as an open-source software as per GNU General Public License v3.0. Source code for LUTD-Base can be accessed from https://github.com/FunctionalUrology/LUTD-Base (accessed on 22 January 2024).

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
