# Peer review of "Machine Learning-Based Classification of Transcriptome Signatures of Non-Ulcerative Bladder Pain Syndrome"

_ijms, 2024, doi:10.3390/ijms25031568_

Round 1

Reviewer 1 Report

Comments and Suggestions for Authors

Authors should be congratulated for their work. The methodology developed to validate the study is polished. Moreover, the topic is worth to be explored due to the economic burden and the mental health-related consequences. Several previous works addressed the topic with different methodologies (PMID= 34559920, 29894774, 22658505). Indeed, the LUTS associated with bladder pain syndrome (BPS) or detrusor overactivity (DO) could cause excessive daytime sleepiness, regardless of gender and clinical features, impairing the psychological balance (PMID= 38002580). All these aspects should be acknowledged in the current manuscript. However, the paper results are well-written and easily readable. The manuscript indeed validates the NGS differences between DO and BPS and applies machine learning to develop a kind of technology to discern biomarkers between BPS and DO. The only thing that I will fix is the caption of some tables and figures that are confusing and short. For instance, Table 1 and Figure 5. Firstly, the caption is too short. It needed an improvement. Moreover, the figure 5  caption should contextualize the aim of the model proposed. 

Author Response

We are thankful to the Reviewers for their insightful comments, and incorporated their suggestions into the text. The detailed replies are below:

Reviewer 1

Authors should be congratulated for their work. The methodology developed to validate the study is polished. Moreover, the topic is worth to be explored due to the economic burden and the mental health-related consequences. Several previous works addressed the topic with different methodologies (PMID= 34559920, 29894774, 22658505). Indeed, the LUTS associated with bladder pain syndrome (BPS) or detrusor overactivity (DO) could cause excessive daytime sleepiness, regardless of gender and clinical features, impairing the psychological balance (PMID= 38002580). All these aspects should be acknowledged in the current manuscript.

Requested citations were incorporated

However, the paper results are well-written and easily readable. The manuscript indeed validates the NGS differences between DO and BPS and applies machine learning to develop a kind of technology to discern biomarkers between BPS and DO. The only thing that I will fix is the caption of some tables and figures that are confusing and short. For instance, Table 1 and Figure 5. Firstly, the caption is too short. It needed an improvement. Moreover, the figure 5 caption should contextualize the aim of the model proposed. 

Table and figure captions were modified as recommended.

Reviewer 2 Report

Comments and Suggestions for Authors

The authors have used machine learning-based approach to analyse bladder biopsy RNA samples with the aim to discern potential gene signature that can help to diagnose non-ulcerative bladder pain syndrome (BPS). Biopsies from BPS patients (n=6), from overactive bladder with detrusor overactivity (DO) patients (n=6) and from asymptomatic control patients (n=6) were collected and subsequently analysed. State of the art next generation sequencing techniques and advanced statistical analysis were used to quantitate RNA expression levels and statistical significance of the findings.

Initially, 13 candidate genes were found capable of discerning BPS from DO and controls. This set of genes were validated from somewhat larger patient cohorts (n=64, total) by QPCR. Supervised and unsupervised ML approaches were then applied.  A three-mRNA signature TPPP3, FAT1 and NCALD was found as a robust classifier of for non-ulcerative BPS.

This study has revealed a three-mRNA signature that may allow more precise diagnosis of BPS. It is remarkable achievement in view of such small number of patient samples.

Lines 220-221: D05 missing from the cluster? Based on Fig. 4, D02, D05 and D06 clustered with control samples.

How authors elucidate/clarify why the initial expression pattern for FAT1 in NGS dataset (=downregulation) was not replicated in subsequent unsupervised or supervised machine learning analyses of qPCR data (=upregulation)? mRNA is not a direct measure for functional protein, so how reliable/relevant qPCR is as validation method, please explain.

I am really surprised that none of the hit genes show up as bladder pain phenotypes in the UK Biobank, that consists of 500 000 exome sequenced individuals. The top phenotypes that associate genetically with FAT1 are prostate disease, pelvic inflammatory disease, hydrocephalus, joint disease, gastroenteritis, rhinitis and tinnitus. Pelvic inflammatory disease may perhaps link FAT1 to BPS. Loss of FAT1 mutations are published to cause glomerulotubular nephropathy (PMID: 26905694). Please, discuss, is FAT1 downregulated or upregulated in BPS and can glomerulotubular nerphropathy drive BPS? Can you rule out glomerulotubular nephropathy as a cause of BPS-like pain?

Author Response

We are thankful to the Reviewers for their insightful comments, and incorporated their suggestions into the text. The detailed replies are below:

Reviewer 2

The authors have used machine learning-based approach to analyse bladder biopsy RNA samples with the aim to discern potential gene signature that can help to diagnose non-ulcerative bladder pain syndrome (BPS). Biopsies from BPS patients (n=6), from overactive bladder with detrusor overactivity (DO) patients (n=6) and from asymptomatic control patients (n=6) were collected and subsequently analysed. State of the art next generation sequencing techniques and advanced statistical analysis were used to quantitate RNA expression levels and statistical significance of the findings.

Initially, 13 candidate genes were found capable of discerning BPS from DO and controls. This set of genes were validated from somewhat larger patient cohorts (n=64, total) by QPCR. Supervised and unsupervised ML approaches were then applied.  A three-mRNA signature TPPP3, FAT1 and NCALD was found as a robust classifier of for non-ulcerative BPS.

This study has revealed a three-mRNA signature that may allow more precise diagnosis of BPS. It is remarkable achievement in view of such small number of patient samples.

  • Lines 220-221: D05 missing from the cluster? Based on Fig. 4, D02, D05 and D06 clustered with control samples.

Thank you for spotting this inconsistency, indeed, it should be Specifically, DO1, DO4, and DO3 clustered together, while DO2, DO5, and DO6 clustered with control samples.” Now corrected in the text.

  • How authors elucidate/clarify why the initial expression pattern for FAT1 in NGS dataset (=downregulation) was not replicated in subsequent unsupervised or supervised machine learning analyses of qPCR data (=upregulation)? mRNA is not a direct measure for functional protein, so how reliable/relevant qPCR is as validation method, please explain.

The features were selected based on having significantly different expression between the groups. The feature selection process disregarded the directionality. If anything, this discrepancy serves as an illustration of unbiasedness of the pipeline. QPCR validation of NGS data, particularly in an unrelated and/or larger population is not without its pitfalls. In general, about 85% of the genes showed consistent results between RNA-sequencing and qPCR data re regulation direction (PMID: 28484260). In our case, QPCR was carried out in a larger independent cohort of patients (n = 28 vs. n = 6 in NGS), and although 11 tested genes showed similar regulation, the observed log2FCs were usually different. However, FAT1 and NRXN3 were up-regulated in the larger cohort based on QPCR compared to controls (also a larger selection). Based on the bigger sample size used in QPCR validation we believe that actually there is an upregulation of FAT1 in BPS. We have included this explanation in the text.

  • I am really surprised that none of the hit genes show up as bladder pain phenotypes in the UK Biobank, that consists of 500 000 exome sequenced individuals. The top phenotypes that associate genetically with FAT1 are prostate disease, pelvic inflammatory disease, hydrocephalus, joint disease, gastroenteritis, rhinitis and tinnitus. Pelvic inflammatory disease may perhaps link FAT1 to BPS. Loss of FAT1 mutations are published to cause glomerulotubular nephropathy (PMID: 26905694). Please, discuss, is FAT1 downregulated or upregulated in BPS and can glomerulotubular nerphropathy drive BPS? Can you rule out glomerulotubular nephropathy as a cause of BPS-like pain?

Thanks a lot for pointing us towards the biobank dataset. We have registered, but unfortunately, we cannot access the biobank to see if there is any BPS-relevant patient sample NGS datasets there. While working on this manuscript, we requested access to the other, albeit few, existing NGS datasets of IC/BPS, which are surprisingly not in public domain, in order to test our signature on yet another patients’ cohort. We failed to get access… therefore this important part of validation is missing.

We commented on the potential role of FAT1 in bladder remodelling. There was no link to kidney disease in our patients with BPS. We discuss this now in the paper.

“TPPP3, FAT1 and NCALD genes are up regulated in BPS (larger patient cohort, QPCR validation), and encode proteins detected in both human and mouse bladders. Tubulin polymerization promoting protein family member 3 (TPPP3) facilitates microtubule assembly [32] and plays a critical role in cell mitosis, one of the main activated processes according to GO ORA of the BPS dataset. Up-regulation of atypical cadherin Fat1 is an important regulator of neuromuscular morphogenesis [33], and its expression in vascular smooth muscle cells increases after injury [34]. Previously, loss of FAT1 has been linked to glomerular nephropathy [35], however, although it was downregulated in the NGS study, its levels were significantly higher than controls in a larger patients’ cohort, ruling out a potential link between the dysfunctions as none of the recruited patients showed kidney deficiency. Interestingly, FAT1 promoted expression of FAT1 was found to promote the expression of pro-inflammatory mediators and TGF-beta [36], potentially contributing to the bladder remodelling in BPS”.